# Contrasting the potential benefits of early invasive coronary angiography in acute and chronic myocardial injury patterns

Joanne Eng-Frost[1,2]☯*, Simon Rocheleau[2☯], Kristina Lambrakis[1,2], Ehsan Khan[1,2], Anke van den Merkhof[1,3], Cynthia Papendick[4], Sam Lehman[1,2], Brian Chiang[2], Naomi Wattchow[2], Simon Steele[2], Scott Lorensini[2], Michael McCann[5], Kate George[2], Julian Vaile[1,2], Carmine De Pasquale[1,2], John French[6], Derek Chew[1,2☯]

1 College of Medicine & Public Health, Flinders University of South Australia, Adelaide, Australia, 2 Department of Cardiovascular Medicine, Flinders Medical Centre, Adelaide, Australia, 3 Faculty of Medical Sciences, University of Groningen, Groningen, Netherlands, 4 School of Medicine, University of Adelaide, Adelaide, Australia, 5 Department of Cardiology, Fiona Stanley Hospital, Perth, Australia, 6 Department of Cardiology, Liverpool Hospital, Sydney, Australia

☯ These authors contributed equally to this work.
* joanne.eng-frost@sa.gov.au

**Data Availability Statement:** All relevant data are within the paper and its Supporting Information files.

## Abstract

### Background

In cases of evolving myocardial injury not definitively attributed to coronary ischaemia precipitated by plaque rupture, referral for invasive coronary angiography (ICA) may be influenced by observed troponin profiles. We sought to explore association between early ICA and elevated high-sensitivity troponin T (hs-cTnT) concentrations with and without dynamic changes, to examine if there may be a hs-cTnT threshold associated with benefit from an initial ICA strategy.

### Methods

Using published studies (hs-cTnT study n = 1937, RAPID-TnT study n = 3270) and the Fourth Universal Definition of Myocardial Infarction (MI), index presentations of patients with hs-cTnT concentrations 5-14ng/L were classified as 'non-elevated' (NE). Hs-cTnT greater than upper reference limit (14ng/L) were classified as 'elevated hs-cTnT with dynamic change' (encompassing acute myocardial injury, Type 1 MI, and Type 2 MI), or 'non-dynamic hs-cTnT elevation' (chronic myocardial injury). Patients with hs-cTnT <5ng/L and/or eGFR<15mmol/L/1.73m$^2$ were excluded. ICA was performed within 30 days of admission. Primary outcome was defined as composite endpoint of death, MI, or unstable angina at 12 months.

### Results

Altogether, 3620 patients comprising 837 (23.1%) with non-dynamic hs-cTnT elevations and 332 (9.2%) with dynamic hs-cTnT elevations were included. Primary outcome was significantly higher with dynamic and non-dynamic hs-cTnT elevations (Dynamic: HR: 4.13

**Funding:** This work is based on two previously published studies (hsTnT and RAPIDTnT). Both studies received funding from the National Health and Medical Research Council (hsTnTstudy award number APP1024008; RAPIDTnT study award number APP1124471), and Roche Diagnostics (unrestricted grant). The funders had no role in study design, data collection and analysis, decision to publish, or preparation of the manuscript.

**Competing interests:** The authors have declared that no competing interests exist.

95%CI:2.92–5.82; p<0.001 Non-dynamic: HR: 2.39 95% confidence interval [CI]:1.74–3.28, p<0.001). Hs-cTnT thresholds where benefit from initial ICA strategy appeared to emerge was observed at 110ng/L and 50ng/L in dynamic and non-dynamic elevations, respectively.

## Conclusion

Early ICA appears to portend benefit in hs-cTnT elevations with and without dynamic changes, and at lower hs-cTnT threshold in non-dynamic hs-cTnT elevation. Differences compel further investigation.

## Introduction

The clinical utility of cardiac troponins (cTnT) has traditionally been its role in identifying acute myocardial infarction (MI) due to unstable atherosclerotic coronary plaques (Type 1 MI). Establishing a diagnosis of Type 1 MI, defined as a dynamic troponin profile with at least one troponin concentrations greater than the 99th percentile of the upper reference range (URL) in conjunction with corresponding clinical, electrocardiographic, or imaging evidence of coronary territory myocardial ischaemia, is necessary in identifying a cohort of patients in whom a strategy of early invasive coronary angiography (ICA), with subsequent coronary revascularization where possible, is associated with a reduction in recurrent ischaemic events [1].

The advent of high-sensitivity troponin (hs-cTnT) assays, with lowered detection thresholds, has ushered in a new standard of urgent care by enabling more rapid "rule in / out" assessment of evolving MI in emergency departments (EDs). It has also conversely unmasked an almost overwhelming cohort of patients with elevated troponin profiles defined by the Fourth Universal Definition of MI (4UDMI) as being reflective of myocardial injury which may not be wholly attributable to underlying plaque rupture. Statically elevated troponin patterns (i.e., non-dynamic troponin elevation) represents a group defined as 'chronic myocardial injury' in the 4UDMI. Conversely, troponin elevation with dynamic changes encompasses a large group of presentations defined as Type 1 MI, Type 2 MI, or acute myocardial injury in the 4UDMI. Differentiating dynamic troponin elevations due to unstable plaque rupture (i.e., Type 1 MI) from other subtypes remains challenging in the absence of coronary anatomy assessment, especially when clinical history, electrocardiographic or imaging supporting coronary ischaemia is subtle or ambiguous, and cardiovascular risk factor profiles are similar [2, 3].

Despite the often-presumed absence of unstable plaque rupture in non-Type 1 MI presentations, these presentations carry similarly poor outcomes with high rates of recurrent cardiac events and a 5-year mortality rate approaching 72% [2, 4]. ICA may offer diagnostic and potentially therapeutic advantages, providing both anatomical assessment for the potential presence of unstable coronary plaque, and the opportunity for definitive revascularization. Additionally, the emergence of CT coronary angiography (CTCA) may offer alternative diagnostic approaches which may in turn inform use of secondary prevention therapies.

However, the evidence-base supporting coronary angiography in this context was largely established prior to the use of hs-cTnT assays [5]. Defining the role of early ICA in reducing recurrent ischaemic events in this undifferentiated troponin elevation cohort remains unexplored. We therefore sought to identify; a) whether the anticipated benefits of an initial ICA strategy differ between dynamically and non-dynamically elevated hs-cTnT patterns; and b) if peak hs-cTnT concentration thresholds beyond which initial ICA is beneficial differ between myocardial injury profiles.

## Methods

### Study design and population

Patients were derived from two previously published randomised controlled trials (RCTs; hs-cTnT study n = 1937, RAPID-TnT study n = 3270) from our group. These are described in detail elsewhere [6, 7]. Briefly, these studies deployed patient-level randomisation to 'unmasked' hs-cTnT results within emergency departments (EDs) and explored its impact on subsequent clinical care and outcomes to 12 months. The RAPID-TnT study included clinical guidance based on the European Society of Cardiology 0/1-hour protocol; testing and interpretation remained unguided in the hs-cTnT study. Importantly, since the latter study excluded patients with obvious ischaemic changes on initial ECG, all subsequent care was at the discretion of treating clinicians. Given the similarity in study designs, data was pooled for this analysis.

Hs-cTnT study enrolled patients in 5 metropolitan Adelaide EDs, and was approved by the Central Adelaide Local Health Network (CALHN) Human Research Ethics committee (HREC), Southern Adelaide Clinical (SAC) HREC, and Northern Adelaide Local Health Network (NALHN) HREC. RAPID-TnT study was approved by SAC HREC. All patients provided written informed consent.

### Troponin testing and myocardial injury profile

All hs-cTnT assays were performed using the Roche Diagnostics Elecys Gen 5 Assay (level of detection 5ng/L in RAPID-TnT study and 3ng/L in hs-cTnT study; upper reference limit [URL] 14ng/L), although clinicians were blinded, or 'masked', to results <29ng/L (i.e., values less than 29ng/L were reported as "<29ng/L" on report available to clinicians) for patients randomized to the control arm in both studies. In the hs-cTnT study, troponin testing was performed at first presentation and 3 hours after symptom onset. Further testing was performed at 6 hours on clinical discretion. In RAPID-TnT, troponin testing was performed at presentation, 3 and 6 hours after presentation, unless discharged before these timepoints. The 2018 4UDMI provided master definitions against which hs-cTnT and RAPID-TnT studies were adjudicated [3]. As the hs-cTnT study predated this, all index and re-presentations with troponin elevations were clinically re-adjudicated against 4UDMI criteria and using the same definitions in RAPID-TnT study to ensure consistency. The unmasked hs-cTnT concentration was used for adjudicating all index and outcome events regardless of randomisation arm allocated in the original studies. Peak troponin concentrations were defined as the highest observed troponin concentration recorded within the first 12 hours of hospital arrival, as previous studies have demonstrated appropriate exclusion of MI if hs-cTnT concentrations were less than URL at presentation and symptom onset > 6 hours from presentation, or with serial testing between 6–12 hours from onset in patients presenting early [8]. Hs-cTnT were categorised as 'non-elevated' if serial levels were between 3–14 ng/L, or 'hs-cTnT elevation with dynamic change' if there was at least one troponin result > 14ng/L and serial levels were associated with rise and/or fall of >20%. This group was further subclassified into Type 1 MI, Type 2 MI, or acute myocardial injury utilising 4UDMI definitions. Results were classified as 'non-dynamic hs-cTnT elevations' if at least one hs-cTnT was >14ng/L, and serial levels were associated with <20% overall change. Index classifications were confined to events adjudicated to have commenced before 12 hours of hospital arrival. Events considered to have commenced after 12 hours were deemed outcome events. Furthermore, patients with an undetectable or low troponin <5ng/L and/or significantly impaired renal function (estimated glomerular filtration rate [eGFR] <15mmol/L/1.73m$^2$) were excluded as these patients were very unlikely to be referred for ICA in clinical practice.

## Coronary angiography and clinical outcomes

An invasive strategy was defined as receipt of ICA occurring within 30 days of randomization. Patients were also evaluated for rates of functional testing (exercise stress ECG testing, stress nuclear, stress echocardiography and stress cardiac magnetic resonance), and subsequent revascularisation (percutaneous coronary intervention and coronary artery bypass grafting).

The composite endpoint of all-cause mortality, new or recurrent MI occurring at least 12 hours after index hospitalization to 12 months follow-up was assessed. The composite of all-cause mortality MI and unstable angina was also reported. Recurrent MI was adjudicated as defined above. Clinical events adjudicated as unstable angina required clinical presentation with chest pain and evidence of ischaemia, or >70% stenosis of any coronary artery by visual estimation on coronary angiography, without elevation in the troponin concentration. Individual components of the composite endpoints at 12 months, and rehospitalisation for the following diagnoses: heart failure, atrial and ventricular arrhythmias, stroke, and peripheral vascular disease were also reported.

A Clinical Events Adjudication Committee consisting of independent cardiologists led by a senior cardiologist evaluated all components of the primary and secondary endpoints. Each case was independently adjudicated by two cardiologists guided by a clinical event committee charter outlining criteria required for a particular diagnosis. Adjudicators were blinded to angiography findings. A diagnosis was considered final if there was consensus between adjudicators. In cases of disagreement, further assessment by a third adjudicator or Clinical Event Committee was sought and resolved by majority.

## Statistical analysis

Baseline characteristics were reported as total numbers with proportions and percentages, or medians and interquartile ranges (IQR), and compared by chi-square test and Kruskal-Wallis tests, respectively. Reflecting the clinical uncertainty surrounding the presence or absence of coronary ischaemia prior to coronary investigations, this analysis grouped patients by index presentation with non-elevated, dynamic elevation, or non-dynamic elevation troponin patterns. Late outcomes by index myocardial injury patterns were explored by Cox proportional hazards models adjusting for age in years, gender, eGFR, prior myocardial infarction, prior heart failure, prior coronary artery bypass grafting and prior percutaneous coronary intervention. Kaplan-Meier survival graphs to 12 months by index myocardial injury pattern were plotted.

To explore the interactions between the peak hs-cTnT level observed within 12 hours of hospital presentation, the index hs-cTnT pattern and reductions in death or acute coronary syndrome (ACS) by 12 months, an inverse probability treatment weight for receipt of ICA within 30 days was developed using the above covariates, peak troponin concentration and troponin pattern (C-statistic:0.83). Within the final weighed population, a standardized difference of <10% was observed for all important prognostic covariates (see S1 File). For each acute and chronic injury population, a logistic regression model with death or recurrent ACS by 12-months as the dependent variable, and including peak troponin concentrations, receipt of ICA, an interaction for angiography versus troponin level, as well as age and eGFR was used to explore the relationship between peak troponin concentration and benefit from ICA. The predicted marginal odds ratio for increasing troponin levels was then plotted. All statistical analyses were conducted using STATA 17 (College Station TX, USA) with significance established at p-values < 0.05.

# Results

## Analysis population

We identified 5207 eligible patients. 22 (0.4%) patients were excluded due to stage five chronic kidney disease (CKD) and 1565 (29.8%) patients were excluded due to hs-cTnT < 5ng/L. Of the remaining 3620 patients, 1006 (41.0%) were men, 2451 (67.7%) patients had non-elevated (NE) troponin concentrations, 332 (9.2%) were adjudicated to have elevated hs-cTnT with dynamic changes (acute pattern), and 837 (23.1%) had hs-cTnT profiles which were elevated but non-dynamic (chronic pattern). Classification of the 332 patients with elevated hs-cTnT levels with evidence of dynamic changes (acute pattern) were: 162 (48.8%) Type 1 MI. 42 (12.7%) Type 2 MI, and 128 (38.6%) as acute myocardial injury (Fig 1).

## Cardiovascular risk factors

Patients with chronic myocardial injury were significantly older, with a higher burden of coronary risk factors including diabetes, hyperlipidaemia, and hypertension (Table 1). Whilst patients with dynamic troponin elevations had higher rates of recurrent MI compared to patients with non-elevated troponin profiles, patients with chronic myocardial injury were significantly more likely to

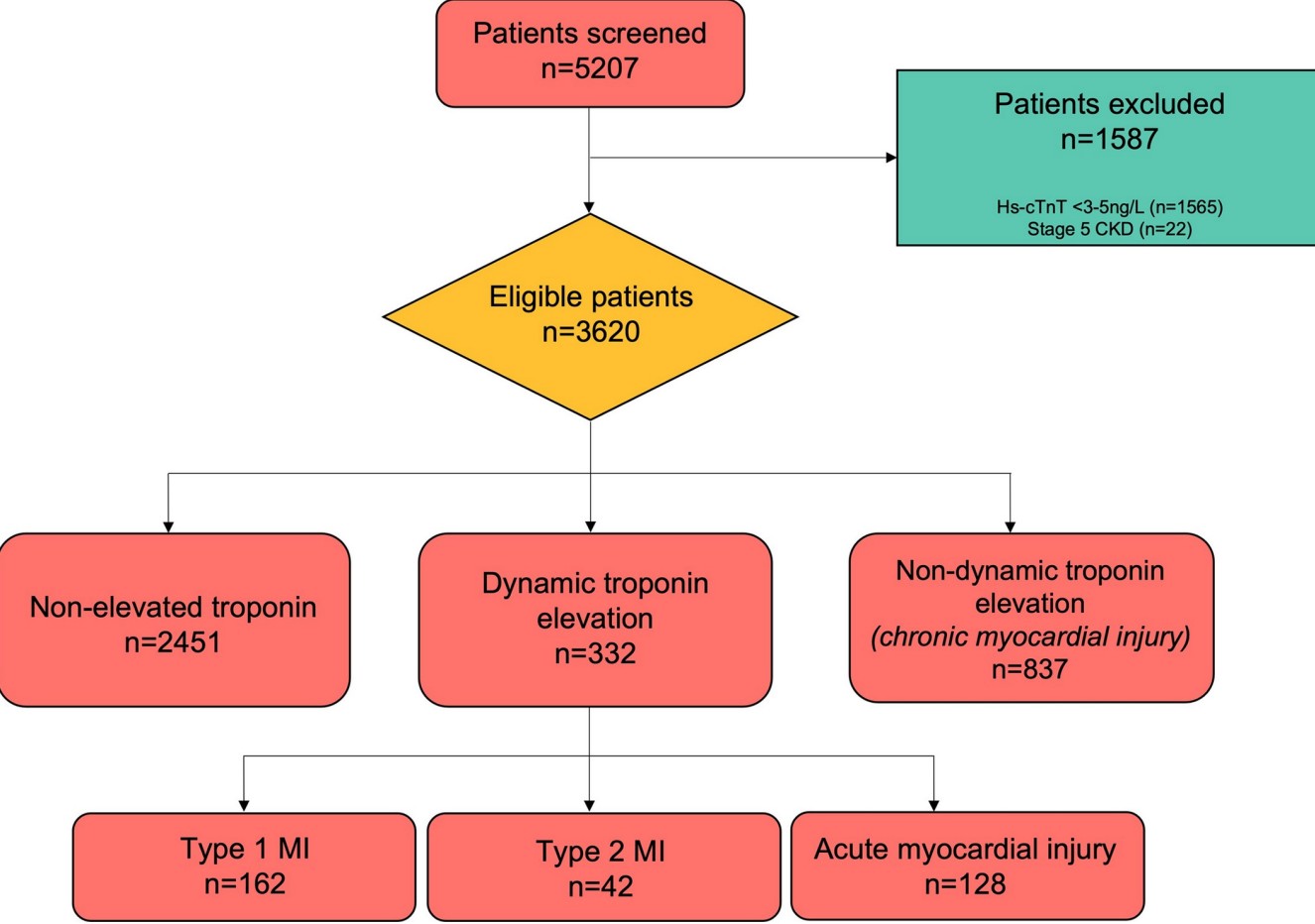

**Fig 1. Screening, eligibility, and clinical adjudication applying the Fourth Universal Definition of MI.** These definitions include dynamic pattern–elevated hs-cTnT with rise and/or fall > 20%; non-dynamic pattern–elevated hs-cTnT with change in troponin < 20%; Type 1 MI–due to coronary plaque rupture; Type 2 MI–due to supply/demand ischaemia; acute myocardial injury–dynamically elevated hs-cTnT due to non-coronary causes; chronic myocardial injury– non-dynamically elevated hs-cTnT due to non-coronary causes).

**Table 1. Baseline characteristics, risk stratification, and outcomes for all patients.**

| | Non-elevated (n = 2451) | Dynamic pattern (n = 332) | Non-dynamic pattern (n = 837) | p-value |
|---|---|---|---|---|
| **Men** | 1006 (41.0%) | 115 (34.6%) | 307 (36.7%) | 0.0141 |
| **Age (years; IQR)** | 61 (51–70) | 68 (57–79) | 77 (66.0–83) | <0.0001 |
| **eGFR (IQR)** | 84.7 (70.1, 96.3) | 77.4 (55.0, 89.3) | 61.4 (44.4, 78.7) | <0.0001 |
| *Comorbidities* | | | | |
| Diabetes | 393 (16.0%) | 81 (24.4%) | 303 (36.2%) | <0.0001 |
| Family history of ischaemic heart disease | 1438 (59.3%) | 199 (60.9%) | 446 (54.1%) | 0.0187 |
| Hyperlipidaemia | 1204 (49.1%) | 187 (56.3%) | 539 (64.4%) | <0.0001 |
| Hypertension | 751 (30.6%) | 145 (43.7%) | 501 (59.9%) | <0.0001 |
| Smoker | 715 (29.2%) | 102 (30.7%) | 189 (22.6%) | 0.0005 |
| Prior acute myocardial infarction | 240 (9.8%) | 59 (17.8%) | 178 (21.3%) | <0.0001 |
| Prior heart failure | 102 (4.2%) | 27 (8.1%) | 179 (21.4%) | <0.0001 |
| Prior atrial fibrillation | 205 (8.4%) | 43 (13.0%) | 211 (25.2%) | <0.0001 |
| Prior chronic obstructive pulmonary disease | 133 (5.4%) | 19 (5.7%) | 144 (17.2%) | <0.0001 |
| Prior cerebrovascular accident | 61 (2.5%) | 11 (3.3%) | 52 (6.2%) | <0.0001 |
| Prior PCI | 223 (9.1%) | 39 (11.7%) | 149 (17.8%) | <0.0001 |
| Prior CABG | 62 (2.5%) | 9 (2.7%) | 81 (9.7%) | <0.0001 |
| *Risk stratification and revascularisation at 30 days* | | | | |
| Exercise stress testing | 402 (16.4%) | 54 (16.3%) | 97 (11.6%) | 0.0033 |
| Diagnostic coronary angiography | 109 (4.4%) | 182 (54.8%) | 94 (11.2%) | <0.0001 |
| Coronary revascularisation | 33 (1.3%) | 102 (30.7%) | 33 (3.9%) | <0.0001 |
| • PCI | 29 (1.2%) | 83 (25.0%) | 27 (3.2%) | <0.0001 |
| • CABG | 4 (0.2%) | 20 (6.0%) | 6 (0.7%) | <0.0001 |
| *Risk stratification and revascularisation at 12 months* | | | | |
| Diagnostic coronary angiography | 226 (9.2%) | 200 (60.2%) | 155 (18.5%) | <0.0001 |
| Coronary revascularisation | 77 (3.1%) | 119 (35.8%) | 57 (6.8%) | <0.0001 |
| • PCI | 62 (2.5%) | 95 (28.6%) | 45 (5.4%) | <0.0001 |
| • CABG | 18 (0.7%) | 28 (8.4%) | 13 (1.6%) | <0.0001 |
| *Outcomes* | | | | |
| All-cause mortality or myocardial infarction | 60 (2.4) | 54 (16.3) | 112 (13.4) | <0.0001 |
| Secondary outcomes | | | | |
| • All-cause mortality | 26 (1.1%) | 28 (8.4%) | 80 (9.6%) | <0.0001 |
| • Myocardial infarction | 34 (1.4%) | 33 (9.9%) | 37 (4.4%) | <0.0001 |
| • Unstable angina | 33 (1.3%) | 7 (2.1%) | 9 (1.1%) | 0.39 |
| • Death, MI, or unstable angina | 89 (3.6%) | 59 (17.8%) | 120 (14.3%) | <0.0001 |
| • Cardiovascular rehospitalisations | 105 (4.3%) | 33 (9.9%) | 121 (14.5%) | <0.0001 |

Values are presented as n (%) or median (IQR).

* n = 2450, + n = 2434, ^ n = 331, # n = 833

Abbreviations: IQR = interquartile range; eGFR = estimated glomerular filtration rate (ml/min/1.73m$^2$); CABG = coronary artery bypass grafting; PCI = percutaneous coronary intervention; ACS = acute coronary syndrome

have a history of prior MI (Non-dynamic: 178, 21.3%; Dynamic: 59, 17.8%; NE: 240, 9.8%; p<0.0001), cerebrovascular accident (Non-dynamic: 52, 6.2%; Dynamic: 11, 3.3%; NE: 61, 2.5%; p<0.0001) and previous coronary revascularisation (Table 1).

## Provision of coronary angiography

Within 30 days, 385 patients (10.6%) underwent ICA. Patients with dynamic changes in elevated hs-cTnT, were significantly more likely to undergo ICA (Dynamic: 182, 54.8%; Non-

**Table 2. Baseline characteristics, risk stratification, and outcomes for patients with dynamic troponin pattern.**

| | No diagnostic coronary angiography (n = 150) | Diagnostic coronary angiography (n = 182) | p-value |
|---|---|---|---|
| **Men** | 58 (38.7) | 57 (31.3%) | 0.16 |
| **Age (years; IQR)** | 75 (59–85) | 65 (56–71) | <0.0001 |
| **eGFR (IQR)** | 74 (45–87) | 79 (63–90) | 0.0033 |
| *Comorbidities* | | | |
| Diabetes | 37 (24.7%) | 44 (24.2%) | 0.92 |
| Family history of ischaemic heart disease | 84 (56.8%) | 115 (64.2%) | 0.17 |
| Hyperlipidaemia | 86 (57.3%) | 101 (55.5%) | 0.74 |
| Hypertension | 74 (49.3%) | 71 (39.0%) | 0.0591 |
| Smoker | 36 (24.0%) | 66 (36.3%) | 0.0159 |
| Prior acute myocardial infarction | 34 (22.7%) | 25 (13.7%) | 0.0341 |
| Prior heart failure | 20 (13.3%) | 7 (3.8%) | 0.0016 |
| Prior atrial fibrillation | 29 (19.3%) | 14 (7.7%) | 0.0017 |
| Prior chronic obstructive pulmonary disease | 11 (7.3%) | 8 (4.4%) | 0.25 |
| Prior cerebrovascular accident | 7 (4.7%) | 4 (2.2%) | 0.21 |
| Prior PCI | 14 (9.3%) | 25 (13.7%) | 0.21 |
| Prior CABG | 2 (1.3%) | 7 (3.8%) | 0.16 |
| *Risk stratification and revascularisation at 30 days* | | | |
| Exercise stress testing | 30 (20.0%) | 24 (13.2%) | 0.0941 |
| Diagnostic coronary angiography | 0 (0.0%) | 182 (100.0%) | <0.0001 |
| Coronary revascularisation[+] | 0 (0.0%) | 102 (56.0%) | <0.0001 |
| • PCI | 0 (0.0%) | 83 (45.6%) | <0.0001 |
| • CABG | 0 (0.0%) | 20 (11.0%) | <0.0001 |
| *Risk stratification and revascularisation at 12 months* | | | |
| Diagnostic coronary angiography | 20 (13.3%) | 182 (100%) | <0.0001 |
| Coronary revascularisation | 8 (5.3%) | 111 (61.0%) | <0.0001 |
| • PCI | 8 (5.3%) | 87 (47.8%) | <0.0001 |
| • CABG | 2 (1.3%) | 26 (14.3%) | <0.0001 |
| *Outcomes* | | | |
| All-cause Death or MI | 30 (20.0%) | 24 (13.2%) | 0.0941 |
| Secondary outcomes | | | |
| • All-cause mortality | 19 (12.7%) | 9 (4.9%) | 0.0117 |
| • Myocardial infarction | 16 (10.7%) | 17 (9.3%) | 0.69 |
| • Unstable angina | 3 (2.0%) | 4 (2.2%) | 0.90 |
| • Death, MI, and unstable angina | 32 (21.3%) | 27 (14.8%) | 0.12 |
| • Cardiovascular rehospitalisations | 22 (14.7%) | 11 (6.0%) | 0.0090 |

Values are presented as n (%) or median (IQR).

* n = 175

[+] = 1 patient received both PCI and CABG during index hospitalisation

Abbreviations: IQR = interquartile range; eGFR = estimated glomerular filtration rate (ml/min/1.73m$^2$); PCI = CABG = coronary artery bypass grafting;

PCI = percutaneous coronary intervention; ACS = acute coronary syndrome

dynamic: 94, 11.2%; NE: 109, 4.4%; p<0.0001; Table 2). Within the acute injury group, patients with Type 1 MI, Type 2 MI and acute injury received angiography in 137 (84.6%), 11 (26.2%) and 34 (26.6%), respectively. Patients with an acute troponin profile were also more likely to receive revascularisation (Dynamic: 102, 30.7%; Non-dynamic: 33, 3.9%; NE: 33, 1.3%; p<0.0001). However, subsequent revascularization occurred predominantly in Type 1 MI patients (92 [56.8%]), compared with 3 (7.1%) in Type 2 MI and 7 (5.5%) in acute injury

**Table 3. Baseline characteristics, risk stratification, and outcomes for patients with non-dynamic troponin pattern (chronic myocardial injury).**

| | No diagnostic coronary angiography (n = 743) | Diagnostic coronary angiography (n = 94) | p-value |
|---|---|---|---|
| **Men** | 279 (37.6%) | 28 (29.8%) | 0.14 |
| **Age (years; IQR)** | 78 (68–84) | 68.0 (58–77) | <0.0001 |
| **eGFR (IQR)** | 60 (43–77) | 68 (53–89) | 0.0017 |
| *Comorbidities* | | | |
| Diabetes | 264 (35.5%) | 39 (41.5%) | 0.26 |
| Family history of ischaemic heart disease | 375 (51.2%) | 71 (77.2%) | <0.0001 |
| Hyperlipidaemia | 478 (64.3%) | 61 (64.9%) | 0.91 |
| Hypertension | 450 (60.6%) | 51 (54.3%) | 0.24 |
| Smoker | 162 (21.6) | 27 (31.0) | 0.0471 |
| Prior acute myocardial infarction | 156 (21.0%) | 22 (23.4%) | 0.59 |
| Prior heart failure | 167 (22.5%) | 12 (12.8%) | 0.0305 |
| Prior atrial fibrillation | 189 (25.4%) | 22 (23.4%) | 0.67 |
| Prior chronic obstructive pulmonary disease | 131 (17.6%) | 13 (13.8%) | 0.36 |
| Prior cerebrovascular accident | 49 (6.6%) | 3 (3.2%) | 0.20 |
| Prior PCI | 127 (17.1%) | 22 (23.4%) | 0.13 |
| Prior CABG | 71 (9.6%) | 10 (10.6%) | 0.74 |
| *Risk stratification and revascularisation at 30 days* | | | |
| Exercise stress testing | 79 (10.6%) | 18 (19.1%) | 0.0151 |
| Diagnostic coronary angiography | 0 (0.0%) | 94 (100.0%) | <0.0001 |
| Coronary revascularisation[+] | 0 (0.0%) | 36 (38.3%) | <0.0001 |
| • PCI | 0 (0.0%) | 31 (33.0%) | <0.0001 |
| • CABG | 0 (0.0%) | 6 (6.4%) | <0.0001 |
| *Risk stratification and revascularisation at 12 months* | | | |
| Diagnostic coronary angiography | 61 (8.2%) | 94 (100.0%) | <0.0001 |
| Coronary revascularisation | 14 (1.9%) | 31 (33.0%) | <0.0001 |
| • PCI | 7 (0.9%) | 6 (6.4%) | <0.0001 |
| • CABG | 21 (2.8%) | 36 (38.3%) | <0.0001 |
| *Outcomes* | | | |
| Primary outcome | 102 (13.7%) | 10 (10.6%) | 0.41 |
| Secondary outcomes | | | |
| • All-cause mortality | 80 (10.8%) | 0 (0.0%) | 0.0008 |
| • Myocardial infarction | 27 (3.6%) | 10 (10.6%) | 0.0019 |
| • Unstable angina | 5 (0.7%) | 4 (4.3%) | 0.0015 |
| • Death, MI, and unstable angina | 107 (14.4%) | 13 (13.8%) | 0.88 |
| Cardiovascular rehospitalisations | 112 (15.1%) | 9 (9.6%) | 0.15 |

Values are presented as n (%) or median (IQR).

[*] n = 746

[+] 1 patient received PCI and CABG during index hospitalisation.

Abbreviations: IQR = interquartile range; eGFR = estimated glomerular filtration rate (ml/min/1.73m$^2$); PCI = CABG = coronary artery bypass grafting;

PCI = percutaneous coronary intervention; ACS = acute coronary syndrome

patients. Patients with non-elevated troponin profile and chronic myocardial injury were less likely to receive early ICA (Dynamic: 182, 54.8%; Non-dynamic: 94, 11.2%, NE: 109, 4.4%; p<0.0001; Table 3). Patients with chronic myocardial injury less likely to undergo functional testing for cardiac risk stratification (Non-dynamic: 97, 11.6%; Dynamic: 54, 16.3%; NE: 402, 16.4%; p = 0.0033) (Table 1).

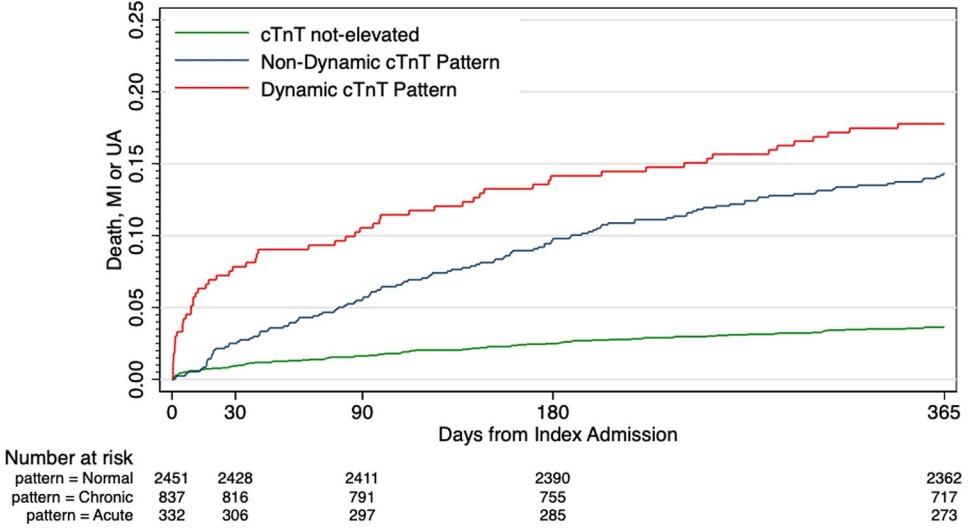

**Fig 2. 12-month all-cause death, myocardial infarction, or unstable angina.**

## Clinical outcomes

Compared to patients with non-elevated troponin profiles at 12-months follow-up, patients with any troponin elevation had higher rates of death, MI, and non-coronary cardiovascular rehospitalisations, with adjusted 12-month hazard ratio (HR) of 4.1 (95% CI: 2.9–5.8) and 2.4 (95% confidence interval [CI]: 1.7–3.73) for troponin elevation with dynamic changes, and troponin elevation without dynamic changes, respectively (Fig 2). All-cause mortality was highest in patients with troponin elevation without dynamic changes (Non-dynamic: 80, 9.6%; Dynamic: 28, 8.4%; NE: 26, 1.1%; p<0.0001).

## Association between coronary angiography and 12-month death or recurrent ACS by hs-cTnT pattern

In patients with dynamic hs-cTnT pattern, a cohort which included Type 1 MI, the receipt of early ICA was associated with a trend towards a lower rate of 12-month death, or MI (ICA: 20, 13.2% vs no ICA: 30, 20.0%; p = 0.094). In this group, there was a lower rate of non-coronary cardiovascular re-hospitalisations with ICA (ICA: 11; 6.0% vs no ICA: 22, 14.7%; p = 0.009). This association was not observed in patients with non-dynamic hs-cTnT pattern who received early ICA, with similar rates of overall death or MI, though there was a higher rate of recurrent MI (ICA: 27, 3.6%; vs no ICA:10; 10.6% p<0.001) but a lower rate of all-cause mortality (ICA: 0, 0.0% vs no ICA: 80, 10.8%; p<0.001). Rates of death, MI, and unstable angina (ICA: 13; 13.8% vs no ICA: 107; 14.4%; p = 0.88) and non-coronary cardiovascular re-hospitalisations (ICA: 9, 9.6% vs no ICA: 112, 15.1%; p = 0.15) were similar irrespective of initial risk stratification strategy.

## Relationship between benefit of angiography and non-dynamic and dynamic hs-cTnT elevations

The predicted 12-month event rates for death or ACS increased as hs-cTnT concentrations increased, in both patients with dynamic and non-dynamic hs-cTnT elevations. An analysis confined to the smaller number of Type 1 MI patients did not demonstrate an interaction between initial observed troponin peak and benefit from angiography (data not shown),

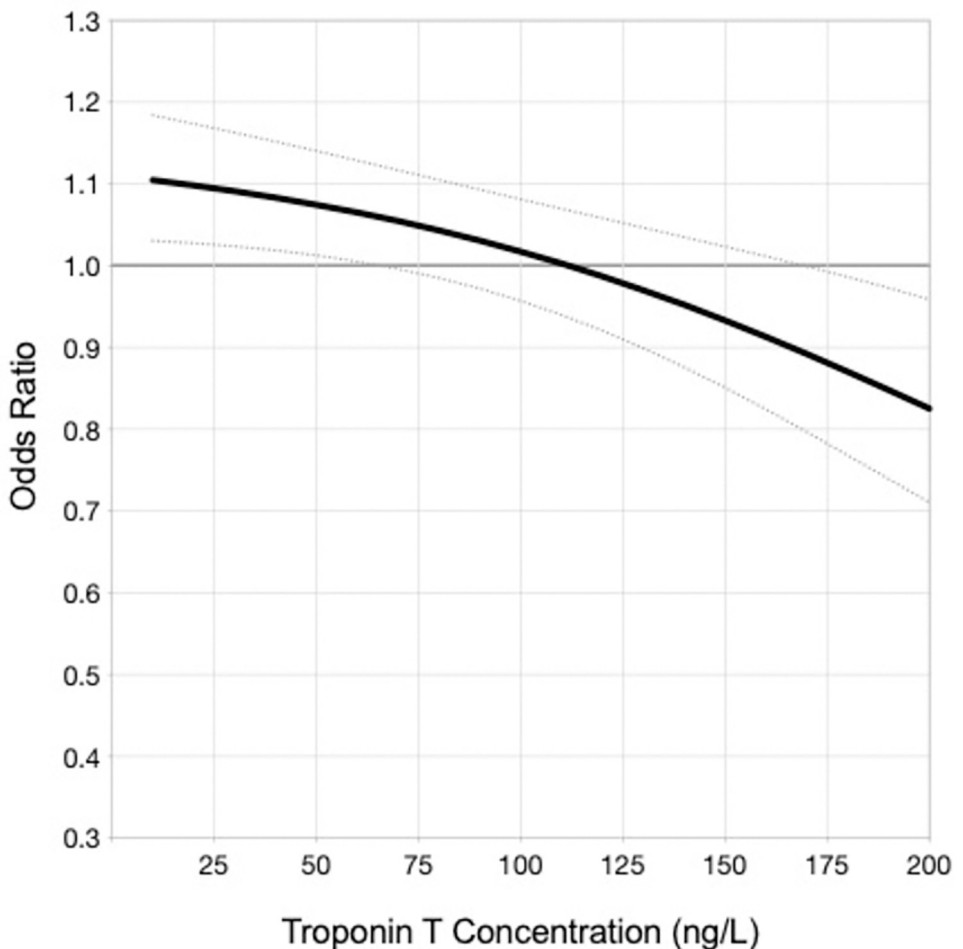

**Fig 3. Predicted odds ratio for death, MI, or unstable angina within 12 months with early ICA in acute myocardial injury.**

however analysis of all patients with an acute injury pattern revealed a threshold of benefit emerging around a hs-cTnT concentration of 110ng/L (Fig 3). Similarly, an initial ICA strategy in patients with chronic myocardial injury also attenuated the rise in 12-month event rates, however benefit appeared to emerge at lower peak hs-cTnT concentration threshold of approximately 50ng/L (Fig 4).

## Discussion

Widespread use of hs-cTnT assays in EDs has unveiled a large cohort of patients with elevated hs-cTnT, both with and without dynamic changes. Further subclassification of these hs-cTnT elevations as Type 1 or 2 MI, acute myocardial injury and chronic myocardial injury as defined in the 4UDMI remains clinically challenging, and appropriate management strategies for non-Type 1 MIs remain poorly defined. In this study of lower risk patients drawn from two randomized clinical trials embedded within ED practice focusing on patients eligible for possible early discharge, more than a third of patients had troponin profiles consistent with non-Type 1 MI, similar to previous reports where chronic myocardial injury accounted for up to 72% of non-Type 1 MIs [4, 9–11]. Consistent with other studies, we demonstrate increased mortality

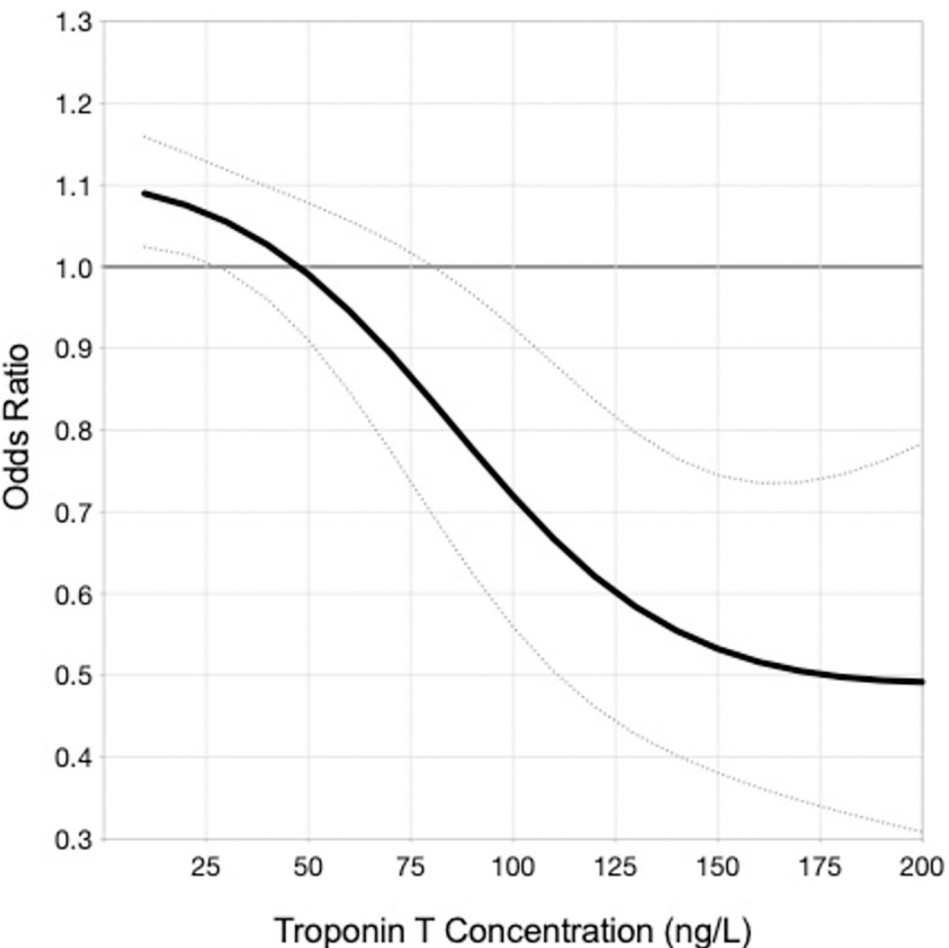

**Fig 4. Predicted odds ratio for death, MI or unstable angina within 12 months with early ICA in non-dynamic hs-cTnT pattern (chronic myocardial injury).**

and recurrent coronary and non-coronary cardiovascular events in patients with elevated hs-cTnT profiles irrespective of associated non-dynamic or dynamic change. This excess cardiovascular risk raises the possibility that a strategy of early ICA with revascularization where possible, may attenuate late coronary ischemic outcomes among patients with elevated hs-cTnT. Whilst exploratory, we observed a lower threshold of potential benefit from an early ICA strategy in patients with non-dynamic hs-cTnT elevation (chronic myocardial injury), compared to patients with dynamic hs-cTnT elevation.

It has become increasingly apparent that the absence of obvious coronary ischaemia in non-Type 1 myocardial injury is not reassuring of innocuity, and in fact carries worrisome implications for short and long-term prognoses [6, 12–14]. Notably, rates of recurrent coronary events, heart failure, arrhythmias, and all-cause mortality in patients with myocardial injury often exceeds corresponding event rates observed in Type 1 MI [6]. Our study confirms that in patients presenting with chest pain, patients with any type of myocardial injury had significantly higher 12-month mortality than patients without evidence of hs-cTnT elevation. Mortality rates were highest in patients with acute myocardial injury, an unsurprising outcome as this population included patients with Type 1 MI. More importantly, we found that patients with chronic myocardial injury had comparably high mortality, with a nearly three-fold

increased risk of death and recurrent cardiovascular events, compared to patients with non-elevated troponin profiles. This may be due to the prevalence of cardio-metabolic risk factors amongst patients with chronic myocardial injury, as evidenced by significantly higher rates of diabetes, hyperlipidaemia, hypertension and previous cardiac or cerebrovascular events in this cohort, compared to both patients with non-elevated troponin profiles, and patients with elevated troponin profiles with dynamic changes.

However, non-cardiac factors also convey equally important prognostic implications that require simultaneous careful consideration [15]. Along with higher rates of traditional cardiovascular risk factors, we found that patients with chronic myocardial injury also had a significantly greater burden of non-coronary risk factors and chronic disease processes including increased age, heart failure, renal impairment, atrial fibrillation, and chronic obstructive pulmonary disease. This is consistent with previous studies demonstrating a higher prevalence of non-coronary comorbidities in patients with non-Type 1 MI troponin profiles [10, 11, 15, 16]. These factors also require simultaneous careful consideration given it has also been suggested that mortality from myocardial injury, especially chronic myocardial injury, is likely predominantly driven by these non-modifiable cardiovascular or non-coronary causes which may not benefit from conventional downstream interventions including ICA and secondary prevention [6, 17]. At present, there are no prospective randomised trials assessing the efficacy of these strategies in myocardial injury. Our exploratory study findings therefore support the need for more prospective research into optimal diagnostic and therapeutic strategies for non-Type 1 MI troponin elevations.

The complex interplay of coronary and non-coronary risk factor burden is further highlighted by the lower recurrent coronary ischaemic event rates observed in patients with chronic myocardial injury who received early coronary angiography. Additionally, we found that the association of coronary angiography with outcome benefit appeared to emerge at a hs-cTnT threshold of 50ng/L in this cohort, in contrast to patients with an acute myocardial injury pattern where the benefit of an initial ICA emerged after a hs-cTnT threshold of 110ng/L. Whilst initially unexpected, this observation may be attributable to a few considerations. First, this finding is consistent with a previous study demonstrating increased long-term all-cause mortality in patients with chronic myocardial injury and hs-cTnT concentrations > 50ng/L, with adjusted mortality risks double that of patients with non-ST elevation MI [18]. Second, we have demonstrated greater cardiovascular risk profiles in patients with chronic myocardial injury, likely predisposing these patients to experiencing symptomatic coronary artery disease and plaque rupture. This combined with a concomitant reduced myocardial reserve due to increased age and non-coronary comorbidities may lead to a propensity for myocardial injury. Third, patients with acute myocardial injury pattern in this study encompassed patients with Type 1 MI as well as patients adjudicated to classifications not due to plaque rupture. This is likely similar to real-world practice, where ICA is used to investigate patients with raised hs-cTnT results that are ultimately attributed to a non- Type 1 MI or coronary ischaemia cause (e.g., myocarditis). Not unexpectedly, such patients are less likely to derive benefit from ICA given the low likelihood of plaque rupture. Fourth, our findings are consistent with previous studies originally exploring an early ICA strategy in an era of troponin assays with lower sensitivity. Thresholds for classification as elevated troponin levels were around 100 ng/L, and patients with troponin levels below this were classified as having "unstable angina" [19–21]. These patients may well have had modest troponin elevations that were below detection threshold of previously available assays that are now detectable with high sensitivity troponin assays. Importantly, routine early ICA was not associated with a benefit in patients with troponin concentrations not considered elevated at the time. Even in patients with confirmed ACS, registry-based analyses have suggested absence of associated benefit with

early ICA when troponin levels were less than 30ng/L [22]. Finally, there has been limited validation of evidence defining the rate of change in elevated hs-cTnT levels which differentiates acute from chronic injury patterns. Accordingly, a relative difference of 20% in the rise and/or fall in troponin conventionally used to differentiate acute and chronic injury patterns may not be relevant in differentiating and prognosticating the potential benefits of ICA in patients presenting with myocardial injury and compels consideration of novel parameters including a narrower window of relative difference or trajectory of difference [23, 24].

Nevertheless, despite the increased mortality risk in patients with chronic myocardial injury and the potential benefits of early ICA in this cohort, we found that these patients were unfortunately less likely to undergo cardiovascular risk stratification with further functional testing or early ICA. The rationale for current practices remains unexplored but is likely due to the greater burden of comorbidities inherent in this population which may make further investigations prohibitive. Prospective randomized validation of a routine coronary investigative strategy among these patients is required given the very common clinical dissonance over their management in acute care settings. The Appropriateness of Coronary Investigation in Myocardial Injury and Type 2 Myocardial Infarction (ACT-2) randomised-controlled trial exploring this in T2MI patients is currently in progress [25; ACTRN 12618000378224].

These observations suggest that more sophisticated methods for matching myocardial injury patterns with optimal coronary investigations are required. Advances in coronary imaging highlights the need to consider the value of non-invasive investigations such as CT coronary angiography [26]. An initial non-invasive coronary assessment approach would uncouple coronary diagnosis from revascularisation, and may potentially be more accessible for patients in rural and regional areas. Utility of a "CTCA first" strategy however requires further prospective randomized studies.

Several limitations should be considered. First, data was drawn from two clinical trials embedded in emergency practice aimed at evaluating the impact of unmasked troponin on investigations and outcomes to facilitate potential early discharges. Consequently, only 32% of patients experienced some form of myocardial injury, precluding assessment of an interaction between early ICA and troponin concentration in each of the diagnostic classifications offered by the 4UDMI. Observations in the acute injury classification should be interpreted cautiously and are merely supportive of prior evidence, with caveats as discussed. It should be noted that clinical, ECG imaging evidence of coronary ischaemia is often ambiguous, therefore requests for inpatient ICA are often strongly influenced by temporal changes in troponin. This analysis has sought to reflect such real-world practices. Second, although this study dataset utilises adjudicated prospective RCT data, it is an observational study which also included patients managed according to troponin results; half of these results were masked to the clinician when initial hs-cTnT levels were <29ng/L. Furthermore, RCTs included in our report utilised the hs-cTnT assays; findings may not be directly translatable to troponin I levels measured with high-sensitivity troponin I assays. These perceived limitations however conversely enhance the utility of the dataset in addressing this question, as this limited information contributed to heterogeneity and clinician autonomy in decisions to refer for ICA. More refined techniques to optimally identify patients who may benefit from an initial ICA approach among the many patients now recognized as having myocardial injury in the context of ED presentations requires further development and prospective research.

Redefining troponin profiles to encompass evolving myocardial injury has become essential with the widespread deployment of increasingly sensitive troponin assays. Optimal investigative strategies for these modest but nonetheless elevated high sensitivity troponin profiles require reconsideration given the increasingly clear associated clinical implications. Early ICA appears to potentially confer benefit in patients with both acute and chronic myocardial injury,

but its limited use and that of emerging diagnostic therapies including CTCA especially in the chronic myocardial injury cohort, requires further prospective randomized investigation.

## Supporting information

**S1 File.**
(DOCX)

**S1 Dataset.**
(XLSX)

## Author Contributions

**Conceptualization:** Cynthia Papendick, Sam Lehman, John French, Derek Chew.

**Data curation:** Kristina Lambrakis, Sam Lehman, John French, Derek Chew.

**Formal analysis:** Sam Lehman, John French, Derek Chew.

**Funding acquisition:** Kristina Lambrakis, Sam Lehman, Derek Chew.

**Investigation:** Kristina Lambrakis, Ehsan Khan, Anke van den Merkhof, Cynthia Papendick, Sam Lehman, John French, Derek Chew.

**Methodology:** Kristina Lambrakis, John French, Derek Chew.

**Supervision:** Sam Lehman, John French, Derek Chew.

**Writing – original draft:** Joanne Eng-Frost, Simon Rocheleau, John French, Derek Chew.

**Writing – review & editing:** Joanne Eng-Frost, Simon Rocheleau, Kristina Lambrakis, Ehsan Khan, Anke van den Merkhof, Cynthia Papendick, Sam Lehman, Brian Chiang, Naomi Wattchow, Simon Steele, Scott Lorensini, Michael McCann, Kate George, Julian Vaile, Carmine De Pasquale, John French, Derek Chew.

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
