## [Decision Letter · Decision Letter 0]

15 Feb 2023

PONE-D-22-32676Contrasting the potential benefits of early invasive coronary angiography in acute and chronic myocardial injury patterns.PLOS ONE

Dear Dr. Eng-Frost,

Thank you for submitting your manuscript to PLOS ONE. After careful consideration, we feel that it has merit but does not fully meet PLOS ONE’s publication criteria as it currently stands. Therefore, we invite you to submit a revised version of the manuscript that addresses the points raised during the review process. You address an important clinical question particularly in the setting of troponin elevation perioperatively in non cardiac surgery.The majority of the troponin elevations are assumed to be due to type 2 MIs and are deemed not to require ICA.What the reader needs from your paper is clear indications when to require ICA in the absence of clear ECG changes. Should they rely on the extent of the troponin elevation or should they require any additional information such as wall motion abnormalities in individuals who had a baseline echo that did not reveal any. Or other clear indications on who might benefit and in whom it would be a waste of resources without any benefit to the patient.

We look forward to receiving your revised manuscript.

Kind regards,

Shukri AlSaif

Academic Editor

PLOS ONE

Journal Requirements:

Reviewers' comments:

Reviewer's Responses to Questions

**Comments to the Author**

1. Is the manuscript technically sound, and do the data support the conclusions?

Reviewer #1: Yes

2. Has the statistical analysis been performed appropriately and rigorously? 

Reviewer #1: Yes

3. Have the authors made all data underlying the findings in their manuscript fully available?

Reviewer #1: Yes

4. Is the manuscript presented in an intelligible fashion and written in standard English?

Reviewer #1: Yes

5. Review Comments to the Author

Reviewer #1: The paper deals with a relevant problem raised by the introduction of hs-cTnC assay. A non negligible number of patients have values above 99th percentile of URL without evident signs of ischemia, with or without dynamic changes. This occurs in patients referred to Emergency Department but also in other settings for example after non cardiac surgery. In both conditions outcome is significantly worse in comparison without hs-cTnC elevation. The authors discuss the role of coronary angiography in different groups of patients . It is not clear indication to angiography in patients without clinical signs of ischemia as well in paragraph "Association between coronary angiography and 12-month death or recurrent ACS by hs-cTnT pattern" they report the sentence "This association was not observed inpatients with chronic myocardial injury who received early ICA, with similar rates of overall death or MI with a higher rate of recurrent myocardial infarction (ICA: 27, 3.6%;

vs no ICA:10; 10.6% p=0.0019) but a lower rate of all-cause mortality (ICA: 0, 0.0% vs no ICA: 80, 10.8%; p=0.0008)" that is not clear . A table or a figure comparing the groups may help the reader . The discussion is long and dispersive and may be shortened and simplified

6. PLOS authors have the option to publish the peer review history of their article (what does this mean?). If published, this will include your full peer review and any attached files.

---

## [Author Response · Author response to Decision Letter 0]

18 Feb 2023

Response to reviewer comments: 

We thank the reviewer for the comment, and recognize the clinical challenges in differentiating Type 2 MI from Type 1 MI based on clinical presentation, and even ECG criteria or wall motion abnormalities on non-invasive imaging. In this analysis, the majority of acute injury cases were Type 1 MI. Importantly, since their differentiation depend on clarity over the pathophysiologic process underlying the myocardial injury (supply/demand ischaemia versus plaque rupture) some may consider assessment of the coronary anatomy/pathology as essential in the diagnostic process.

This analysis takes a different approach. By exploring the profile and peak concentration of troponin T as a determinant of “prognostic benefit” from invasive coronary angiography (as opposed to the use in diagnosis) , we have suggested that there is are threshold values, depending on the profile, below which a benefit from an invasive coronary is not apparent, and therefore alternative strategies for coronary investigation, such as CTCA, may also be a viable strategy. Hence, this analysis suggests that in patients with an acute myocardial injury pattern, regardless of whether the diagnosis is Type 1 MI, Type 2 MI or acute injury, the early invasive strategy among patients with a peak troponin T less than ~100ng/L is unlikely to impact future death or MI. This insight may help clinicians consider alternative investigative approaches to those patients with low level troponin elevations.

Response to editor comments:

We thank the editor for their time. 

1. The manuscript has been reviewed to ensure it meets PLOS ONE's style requirements.

2. The minimum dataset has been fully anonymised, and has been included as "Supporting Information Compressed Zip file" with the latest revision.

3. The ethics statement has been included in the "Methods" section of the manuscript. It includes the name of the Human Research Ethics Committees who approved the study, and all participants provided full written consent.

4. Captions for the supporting information files have been included at the end of the manuscript.

---

## [Decision Letter · Decision Letter 1]

8 Mar 2023

PONE-D-22-32676R1Contrasting the potential benefits of early invasive coronary angiography in acute and chronic myocardial injury patterns.PLOS ONE

Dear Dr. Joanne Eng-Frost,

Thank you for submitting your manuscript to PLOS ONE. After careful consideration, we feel that it has merit but does not fully meet PLOS ONE’s publication criteria as it currently stands. Therefore, we invite you to submit a revised version of the manuscript that addresses the points raised during the review process.

 Please submit your revised manuscript by end of April.  If you will need more time than this to complete your revisions, please reply to this message or contact the journal office at plosone@plos.org. Please include the following items when submitting your revised manuscript:A rebuttal letter that responds to each point raised by the academic editor and reviewer(s). You should upload this letter as a separate file labeled 'Response to Reviewers'.A marked-up copy of your manuscript that highlights changes made to the original version. You should upload this as a separate file labeled 'Revised Manuscript with Track Changes'.An unmarked version of your revised paper without tracked changes. You should upload this as a separate file labeled 'Manuscript'.If applicable, we recommend that you deposit your laboratory protocols in protocols.io to enhance the reproducibility of your results. Protocols.io assigns your protocol its own identifier (DOI) so that it can be cited independently in the future. For instructions see: https://journals.plos.org/plosone/s/submission-guidelines#loc-laboratory-protocols. Additionally, PLOS ONE offers an option for publishing peer-reviewed Lab Protocol articles, which describe protocols hosted on protocols.io. Read more information on sharing protocols at https://plos.org/protocols?utm_medium=editorial-email&utm_source=authorletters&utm_campaign=protocols.

We look forward to receiving your revised manuscript.

Kind regards,

Shukri AlSaif

Academic Editor

PLOS ONE

Journal Requirements:

Reviewers' comments:

Reviewer's Responses to Questions

**Comments to the Author**

1. If the authors have adequately addressed your comments raised in a previous round of review and you feel that this manuscript is now acceptable for publication, you may indicate that here to bypass the “Comments to the Author” section, enter your conflict of interest statement in the “Confidential to Editor” section, and submit your "Accept" recommendation.

Reviewer #1: (No Response)

2. Is the manuscript technically sound, and do the data support the conclusions?

Reviewer #1: Yes

3. Has the statistical analysis been performed appropriately and rigorously? 

Reviewer #1: Yes

4. Have the authors made all data underlying the findings in their manuscript fully available?

Reviewer #1: Yes

5. Is the manuscript presented in an intelligible fashion and written in standard English?

Reviewer #1: Yes

6. Review Comments to the Author

Reviewer #1: The paper is interesting and outlines the difficulties in the interpretation of high hs-TnT levels . In figure 1 I do not understand who is inserted in the third group (acute MI??). In table II angiography has been performed in 182 patients . Overall coronary revascularisation was reported in 83 (45.6%) but thereafter are described PCI 20 (11.0%) and CABG 102 (56.0%) ( a total of 122 patients) . The numbers are not concordant. T We agree that hs TnT is related to long term survival both in patients with chronic or changing troponin levels . Whether patients in RAPID study were included without ECG examination how was diagnosis of type of MI made ? The patients from the two study had ED evaluation for chest pain ? It should be reported in the description of the study. The number of patients with chronic myocardial injury underwent angiography was close to 15% of the whole population and only a small number were studied at 12 months with 3.7% of further revascularization Overall mortality was not different from the other group , the other end points are differently reported in table 2 and 3 ( for example ACS death in table 2 is not reported in table 3, Death, MI, and unstable angina in table 3 ) . A table comparing the two groups may be useful . The paper is long and discussion not easy to follow. Please simplify fo the reader.

7. PLOS authors have the option to publish the peer review history of their article (what does this mean?). If published, this will include your full peer review and any attached files.

Reviewer #1: No

---

## [Author Response · Author response to Decision Letter 1]

28 Apr 2023

Response to reviewer comments

1. In figure 1 I do not understand who is inserted in the third group (acute MI??). 

There were 3 groups of patients included in this study. 

Patients in the “non-elevated troponin group” were defined as patients with serial troponins between 3-14ng/L.

Patients in the “acute myocardial injury group” (or hs-cTnT elevation with dynamic change) were defined as patients with at least 1 troponin result > 14ng/L and serial levels associated with rise and/or fall of > 20%. This group was further subclassified into Type 1 MI, Type 2 MI, or acute myocardial injury, utilising the 4th Universal Definition of MI definitions. 

Patients in the “chronic myocardial injury group” (or non-dynamic hs-cTnT elevation) were defined as patients with at least 1 hs-cTnT result > 14ng/L but serial levels were associated with < 20% overall change. 

Figure 1 has been amended for clarity. 

2. In table II angiography has been performed in 182 patients. Overall coronary revascularisation was reported in 83 (45.6%) but thereafter are described PCI 20 (11.0%) and CABG 102 (56.0%) - a total of 122 patients. The numbers are not concordant.

We thank the reviewer for the question. There is a transcription error in the table that has now been corrected. Among those with a dynamic pattern, 102 received some form of revasculrisation. Of the 102, 83 patients received PCI, whilst 20 received CABG – 1 person received both PCI and CABG. 

3. Whether patients in RAPID study were included without ECG examination how was diagnosis of type of MI made ? The patients from the two study had ED evaluation for chest pain ? It should be reported in the description of the study. 

The design of RAPID-TnT has previously been published - https://www.sciencedirect.com/science/article/pii/S000287031730145X?casa_token=YJrw8RmRTrUAAAAA:0Xj_t2GLiBT0W2F-TokR042F_WlmyhDb8VLdBZxCQUtxCSMhI-QeI9Did0x_cBhyLcd_tEdeNIQ. 

The inclusion criteria for patients in RAPID TnT were:

- Clinical features of chest pain or suspected ACS as the principal cause for investigation

- Baseline ECG interpreted as not definitive for coronary ischaemia

- Age > 18yo

- Willing to give written informed consent

Patients were then randomised to the 0/1-hour protocol, with the result to determine the patient’s subsequent care. There were 3 possible outcomes:

1. Baseline troponin < 5, or <12ng/L and change in troponin over 1 hour < 3ng/L -> rule-out i.e. discharge to primary care 

2. Baseline troponin > 52ng/L, or change over 1 hour of > 5ng/L -> rule-in i.e. admit to hospital for management of MI

3. Baseline troponin 13-51ng/L and a change <5ng/L over 1 hour, or baseline troponin < 12ng/L and a change of 3-4ng/L over 1 hour -> observe

The design of hs-cTnT has previously been published here. https://www.ahajournals.org/doi/10.1161/CIRCOUTCOMES.115.002488.

Patients presenting to emergency departments with clinical features or suspected ACS (chest pain or overwhelming shortness of breath > 10 minutes at rest < 24 hours from the time of presentation) in whom the treating physician deemed it necessary to measure serum troponin were eligible.

4. The number of patients with chronic myocardial injury underwent angiography was close to 15% of the whole population and only a small number were studied at 12 months with 3.7% of further revascularization Overall mortality was not different from the other group , the other end points are differently reported in table 2 and 3 ( for example ACS death in table 2 is not reported in table 3, Death, MI, and unstable angina in table 3 ) . A table comparing the two groups may be useful. 

We apologise for the confusion. There is a transcription error in the table that has now been corrected and the labels for the clinical outcome have also been corrected. Tables 2 and 3 were intended to represent exactly the same parameters. 

5. The paper is long and discussion not easy to follow. Please simplify for the reader.

We have reviewed the manuscript and have made amendments to enhance clarity.

---

## [Editor Report · Decision Letter 2]

10 May 2023

Contrasting the potential benefits of early invasive coronary angiography in acute and chronic myocardial injury patterns.

PONE-D-22-32676R2

Dear Dr. Eng-Frost,

We’re pleased to inform you that your manuscript has been judged scientifically suitable for publication and will be formally accepted for publication once it meets all outstanding technical requirements.

Kind regards,

Shukri AlSaif

Academic Editor

PLOS ONE

---

## [Editor Report · Acceptance letter]

7 Jun 2023

PONE-D-22-32676R2 

Contrasting the potential benefits of early invasive coronary angiography in acute and chronic myocardial injury patterns. 

Dear Dr. Eng-Frost:

I'm pleased to inform you that your manuscript has been deemed suitable for publication in PLOS ONE. Congratulations! Your manuscript is now with our production department. 

Kind regards, 

on behalf of

Dr. Shukri AlSaif 

Academic Editor

PLOS ONE